# Controllable Nitric Oxide Storage and Release in Cu-BTC: Crystallographic Insights and Bioactivity

**DOI:** 10.3390/ijms23169098

**Published:** 2022-08-13

**Authors:** Do Nam Lee, Yeong Rim Kim, Sohyeon Yang, Ngoc Minh Tran, Bong Joo Park, Su Jung Lee, Youngmee Kim, Hyojong Yoo, Sung-Jin Kim, Jae Ho Shin

**Affiliations:** 1Ingenium College of Liberal Arts (Chemistry), Kwangwoon University, Seoul 01897, Korea; 2Department of Chemistry, Kwangwoon University, Seoul 01897, Korea; 3Nanobio-Energy Materials Center, Department of Chemistry and Nano Science, Ewha Womans University, Seoul 03760, Korea; 4Department of Materials Science and Chemical Engineering, Hanyang University, Ansan-si 15588, Korea; 5Department of Chemistry, University of Sciences, Hue University, Hue City 530000, Vietnam; 6Department of Electrical and Biological Physics, Kwangwoon University, Seoul 01897, Korea

**Keywords:** nitric oxide, drug delivery, MOFs, antibacterial activity

## Abstract

Crystalline metal–organic frameworks (MOFs) are extensively used in areas such as gas storage and small-molecule drug delivery. Although Cu-BTC (**1**, MOF-199, BTC: benzene-1,3,5-tricarboxylate) has versatile applications, its NO storage and release characteristics are not amenable to therapeutic usage. In this work, micro-sized Cu-BTC was prepared solvothermally and then processed by ball-milling to prepare nano-sized Cu-BTC (**2**). The NO storage and release properties of the micro- and nano-sized Cu-BTC MOFs were morphology dependent. Control of the hydration degree and morphology of the NO delivery vehicle improved the NO release characteristics significantly. In particular, the nano-sized NO-loaded Cu-BTC (NO⊂nano-Cu-BTC, **4**) released NO at 1.81 µmol·mg^−1^ in 1.2 h in PBS, which meets the requirements for clinical usage. The solid-state structural formula of NO⊂Cu-BTC was successfully determined to be [CuC_6_H_2_O_5_]·(NO)_0.167_ through single-crystal X-ray diffraction, suggesting no structural changes in Cu-BTC upon the intercalation of 0.167 equivalents of NO within the pores of Cu-BTC after NO loading. The structure of Cu-BTC was also stably maintained after NO release. NO⊂Cu-BTC exhibited significant antibacterial activity against six bacterial strains, including Gram-negative and positive bacteria. NO⊂Cu-BTC could be utilized as a hybrid NO donor to explore the synergistic effects of the known antibacterial properties of Cu-BTC.

## 1. Introduction

Nitric oxide (NO) in the environment is mainly derived from natural sources and internal combustion engines. It is typically considered an air pollutant and causes smog and strongly acidic rain. However, since Furchgott et al. reported that NO acts as the endothelium-derived relaxation factor, NO has also been identified as an important physiological and pathological signaling material responsible for regulating the cardiovascular and nervous systems as well as the immune response [1,2,3,4,5,6]. The deficiency of NO produced endogenously from l-arginine by NO synthase (NOS) causes various diseases, which has driven investigations into the exogenous delivery of NO for therapeutic applications. Because NO is a highly reactive radical and gains an electron through covalent bonding, hydrogen bonding, or coordinative bonding [7,8,9,10], several types of NO donors have been explored for efficient NO delivery. These include simple nitrosyl metal complexes of organic nitrates/nitrites, macromolecular scaffolds containing nitrosamines, *N*-diazeniumdiolates (NONOates), and *S*-nitrosothiols [11,12,13,14,15].

Metal–organic frameworks (MOFs) have been used as attractive organic–inorganic hybrid materials that have high porosity and crystallinity and enable the facile tuning of structural composition. Consequently, MOFs have been used as sensors, applied to catalysis and gas sorption and separation, and employed in medicinal applications [16,17,18,19,20,21,22,23]. Moreover, MOFs have been considered the best candidates for controlled NO release as they can physically capture NO in their inner pores, bond with NO via their open metal sites (OMSs), or amine functional group of the linkers comprising the frameworks [24,25,26,27,28,29,30,31,32,33]. Cu-BTC, formulated as Cu_3_(BTC)_2_ (MOF-199, copper(II)-benzene-1,3,5-tricarboxylate), is one of the most studied MOFs and is commonly used as a sensor or for gas sorption, gas storage, and catalysis [34,35,36,37,38,39]. For example, Xiao et al. investigated Cu-BTC as an NO donor due to its high crystallinity and varied porosity and reported that this material could store as much as 9 mmol·mg ^−1^ of NO at 196 K but released only 1 nmol·mg^−1^—a very small fraction of the loading—upon exposure to a stream of wet nitrogen gas. However, this irreversible release was detrimental to the application of Cu-BTC for therapeutic purposes [28]. This result also suggested that Cu-BTC predominantly physically absorbs NO in its inner pores, with some NO being partially chemically adsorbed onto its OMSs. To address this limitation, other isostructural MOFs incorporating secondary amine functionalities were developed for enhancing NO release up to 0.51 µmol·mg^−1^ [11]. Subsequently, Co-CPO-27 and Ni-CPO-27 linked to the 2,5-dihydroxyterephthalic acid ligand were reported as the best MOF donors, which absorbed 6.5 and 7.0 µmol·mg^−1^ of NO, respectively. While they successfully released the absorbed NO completely, their poor biocompatibility limited their applications [40]. The bonding of NO to MOFs has been typically characterized using FTIR spectroscopy, NMR spectroscopy, or theoretical simulations, which revealed the formation of nitrosyl complexes between NO and the OMSs or NONOates derived from secondary amines [31,32,40,41,42,43]. However, there are no reports on the characterization of the NO loading modes in the solid state by X-ray crystallography, mainly due to the rapid release of NO in the presence of water vapor and the challenges related to low-temperature measurements. To improve the NO storage/release properties of MOFs, we aimed to understand the bonding characteristics of NO loaded on MOFs by single-crystal X-ray crystallography, which is expected to be significantly different from previous studies [28,31].

Herein, for improving the biocompatibility and NO capacity, nano-sized Cu-BTC was fabricated via a convenient and straightforward ball-milling process of micro-sized Cu-BTC. The hydration degree of Cu-BTC was controlled to enable the coordination of sufficient water molecules to the OMSs. However, the morphology of Cu-BTC can be designed to load NO efficiently in the empty pores by the facile removal of the guest water molecules by activation. Highly improved NO storage/release and antibacterial activities against six bacterial strains, including Gram-positive 

Gram-negative bacteria, are discussed.

## 2. Results and Discussion

### 2.1. Preparation of Cu-BTC and NO⊂Cu-BTCs

To investigate the NO storage/release properties and antibacterial activities of Cu-BTC, micro-sized Cu-BTC (**1**) was prepared. A previously reported solvothermal method, after slight modification, was used for the reaction of copper(II) nitrate and benzene-1,3,5-tricarboxylic acid in a mixture of ethanol, deionized water, and DMF [44]. The resulting compound **1** was further processed by ball-milling to afford nano-sized Cu-BTC (**2**) (Appendix A). The two NO-loaded Cu-BTC systems, NO⊂micro-Cu-BTC (**3**; NO was loaded on **1**) and NO⊂nano-Cu-BTC (**4**; NO was loaded on **2**) were obtained by charging 10 atm of NO onto activated **1** and **2**, respectively, at 25 °C for three days. The obtained Cu-BTC and NO-loaded Cu-BTC were characterized by X-ray crystallography, PXRD, SEM, Brunauer–Emmett–Teller (BET) measurements, FTIR spectroscopy, and TGA.

### 2.2. X-ray Crystallography and PXRD

The structures of all Cu-BTC MOFs before and after NO gas loading were determined by X-ray crystallography and PXRD. A single crystal with 0.02 × 0.08 × 0.09 mm^3^ dimensions was selected for XRD analysis. Upon the capture of NO gas in **1**, its cubic *Fm-3m* space group (Table 1) and the original framework remained unchanged (Figure 1). X-ray crystallography revealed the moiety formula of **3** to be CuC_6_H_2_O_5_]·(NO)_0.167_. While the occupancies of NO were fixed to obtain the best fit with the largest residual peaks, the actual occupancies should be larger than the fixed occupancies. The single-crystal structure of **3** showed that the remaining water molecules were coordinated to the OMSs. NO interacts through weak hydrogen bonding between the hydrogen atom of the coordinated water and the nitrogen atom of NO (Cu–O–H···N–O, 3.308 Å). Weak interactions between the nitrogen atom of NO and the carboxylate oxygen atoms of the BTC linkers (Cu–O–C–O···NO, 3.999 Å) were also observed (Figure 2 and Appendix A).

Notably, this is the first report of a single-crystal X-ray structure that clearly shows the NO loading structure, which comprises NO interacting with the coordinated water molecules and the carboxylate groups of the BTC linkers. The NO loading mode of 3 in this study was significantly different from previous results that characterized the adsorption of NO to the OMSs as chemisorption [28,31]. Figure 3 shows the XRD patterns of the simulated and NO-loaded Cu-BTC MOFs of different sizes. All XRD patterns coincided well with the simulated pattern. The main peaks at 6.82°, 9.64°, 11.76°, and 13.57° corresponding (2 0 0), (2 2 0), (2 2 2), and (4 0 0) of the Cu-BTC pattern were maintained in nano-Cu-BTC and micro-Cu-BTC. The (2 0 0) peak was not shown in NO loaded Cu-BTC MOFs. Particularly, the PXRD of **4** obtained after storing NO for 1 month did not show any structural changes, and the original framework of Cu-BTC remained unchanged. These results indicate that the robust structure of Cu-BTC was stably maintained after storing NO for a long time, which could enable it to perform the reversible NO release mechanism.

### 2.3. SEM, BET Measurements, FTIR, and TGA

The morphologies of the Cu-BTCs before and after loading NO were imaged using SEM. As shown in Figure 4a, truncated octahedral crystals larger than 10 µm in length were observed, which is in good agreement with the previous reports [45,46,47]. In contrast to the micro-sized crystal, **2** was seen as an irregular lump comprising an agglomerated powder with sizes in the range of less than 500 nm to greater than 1 µm (Figure 4b). This may be attributed to the loss of crystallinity during the ball-milling process, which could cause the transformation to the agglomerated powder [48]. The morphologies of the Cu-BTCs were well maintained after loading NO (Figure 4c,d).

In addition, the surface properties of **1** and **2** were analyzed by N_2_ sorption measurements at 77 K. A high BET surface area of 799 m^2^·g^−1^ was achieved by **1**, and its nitrogen sorption (Figure 5a) showed a characteristic type-I isotherm [49,50]. The increased adsorption uptakes at low relative pressures (P/P_0_ ≤ 0.1) are due to the presence of micro pores, whereas a hysteresis at high relative pressures (0.3 ≤ P/P_0_ ≤ 0.9) indicates the existence of textural meso or macro pores, formed as a result of the specific crystal packing. The large surface area of **1** might have originated from its high crystallinity. In contrast, **2** exhibited a much lower BET surface area (27 m^2^·g^−1^) than **1**. Further, **2** exhibited a characteristic type-II isotherm, proving that it is composed of both mono and multilayers (Figure 5c) with mixed micro pore diameters ranging from 0.4 to 2.0 nm (Figure 5d) [50]. The smaller surface area of the nano-sized crystal of **2** can be attributed to the agglomeration of nano powder and/or the defects of its inner pores formed during the ball-milling process [51]. Furthermore, the vertical nature of the N_2_ adsorption isotherms of **1** at low pressures supports the hypothesis that these adsorption/desorption properties are derived from the strong adsorption on the surface and the micro pore filling. The plot of **2** suggests that intra-agglomerate voids arise from the presence of some meso and macro pores formed by multilayers in the samples [52,53]. We have summarized the nitrogen sorption properties of the micro-sized Cu-BTC and the nano-sized Cu-BTC on Appendix A. These results match well with the SEM images shown in Figure 4b, showing the lump shape composed of the agglomerated powder.

Cu-BTC was degassed at 300 °C under 1 × 10^−4^ mbar of pressure using a vacuum to ensure complete dehydration. The loading of NO onto this MOF was confirmed based on the appearance of stretching peaks in the 1890–1000 cm^−1^ region of its FTIR spectrum, which is attributed to the formation of the nitrosyl complex by the coordination of NO on the reactive OMSs [28]. Further, the FTIR spectra of **3** and **4** were scanned at various times after the NO was released (Figure 6 and Figure 7). The Cu-BTC MOFs before and after loading NO showed weak bands at 488 and 721 cm^−1^, which are attributed to the bending and stretching modes of Cu–O, respectively. The higher intensity absorption peaks at 1368, 1445, and 1640 cm^−1^ are assigned to the stretching modes of the carboxylate moiety of BTC; specifically, the stretching mode of C–O, and the asymmetric and symmetric stretching modes of C=O, respectively [34,35,54]. However, in this study, a new peak representing the coordination of NO to Cu in the 1890–1900 cm^−1^ range did not appear after NO loading. Rather, a strong, broad O–H stretching band at ~3400 cm^−1^ was observed, which corresponds to the water molecules coordinating with the OMSs. This evidence of water coordinating with Cu agrees with the NO loading mode determined by the crystallographic structure of **3**.

TGA was carried out to analyze the amount of water molecules coordinated to the OMSs and the amount of guest water molecules captured in the inner pores or on the surface, depending on the activation. When as-prepared **1** and **2** were heated up to 800 °C under an inert atmosphere, an initial weight loss occurred up to 150 °C, which was due to the desorption of the physically absorbed water (Figure 8). Then, slow weight loss caused by the desorption of the water molecules coordinating with the OMSs continued up to 350 °C. The final prominent weight loss step appeared at 350 °C in both the samples and was attributed to the decomposition of the BTC linker (approximately 38% weight loss). The largest difference in the initial weight loss ratio depending on the morphologies of **1** and **2** was 9%. Furthermore, the hydration of **2** could be reduced from 27% to 20.6% upon activation at 150 °C for 24 h (Figure 8b). The hydration of Cu-BTC was adequately lowered to 20.6% (150 °C) by controlling the morphology and activation. We attempted to investigate the hydration effect on NO storage and release.

### 2.4. NO Release from NO⊂Cu-BTC

The amount of NO released from NO⊂Cu-BTC was estimated from the intensity of the chemiluminescence of excited NO_2_ that was produced from the reaction of the NO with ozone at the outlet stream of the PBS solution at 37 °C. Cu-BTC is a well-known MOF that adsorbs 3 µmol·mg^−1^ of NO at room temperature; however, 2.21 µmol·mg^−1^ of NO was strongly chemisorbed onto the OMSs of the MOF, and only a minor fraction of the chemisorbed NO, less than 0.02%, was released upon exposure to water or wet gas [28].

The chemisorption of NO onto the OMSs inhibits its release sufficiently, thus inhibiting the therapeutic applications of our MOFs. Thus, we attempted to control the chemisorption of NO onto the OMSs by adjusting the hydration degree, NO charging pressure, and the morphologies of the sample to enhance the NO release ability of Cu-BTC.

Figure 9 shows the NO release tendency of NO⊂micro-Cu-BTC **3** as a function of the NO charging pressure in a range of 2 to 12 atm. The total amount of NO released (t[NO]) increased up to 0.34 µmol·mg^−1^ with increasing charging pressure. Further, to investigate the effect of dehydration on NO release, unactivated (i.e., non-dehydrated) Cu-BTC and Cu-BTC activated (i.e., dehydrated) at 150 °C for 24 h were examined. The t[NO] value of dehydrated **3** increased approximately two-fold, from 0.34 (non-dehydrated) to 0.68 µmol·mg^−1^, and the duration of NO release (*t*_d_) was longer than that of the non-dehydrated sample. Furthermore, the t[NO] value of dehydrated **4** increased significantly from 0.25 (non-dehydrated) to 1.81 µmol·mg^−1^, and the maximum flux of NO release ([NO]_m_) increased proportionally from 4.78 (non-dehydrated) to 52.7 ppm·mg^−1^ (Figure 10). The notable difference between dehydrated **3** and **4** was attributable to the different surface areas and porosities, which were derived from their morphologies (Table 2). These results confirm that more regular micro pores exist on **3** than on **4**, as shown on Figure 5. As the guest water molecules trapped in these regular inner pores cannot be easily released, the loading of NO on the micro pores of **3** is more difficult compared to that of **4**. As a result, NO can be adsorbed in more empty pores and on the surface of **4,** and t[NO] reaches to 1.81 µmol·mg^−1^ upon exposure to water, which is sufficiently high for therapeutic applications [55].

### 2.5. Antibacterial Properties

Therapeutic applications involving NO are popular in various areas, such as immunology, studies on antibacterial and anticancer agents, as well as wound healing. However, NO sometimes exhibits a dual effect depending on its concentration, flux, and duration when used as an antibacterial agent [6,55,56,57]. To evaluate the antibacterial activity of our NO-releasing Cu-BTC, an inhibition zone assay based on a modified disk diffusion method was carried out using two Gram-negative strains (*Escherichia coli* (*E. coli* ATCC11775) and *Pseudomonas aeruginosa* (*P. aeruginosa* ATCC9027)), two Gram-positive strains (*Staphylococcus aureus* (*S. aureus* ATCC14458) and *Bacillus cereus* (*B. cereus* ATCC11706)), and two methicillin-resistant *Staphylococcus aureus* strains (MRSA KCCM40510 and clinically isolated MRSA) (Figure 11). The inhibition zone of the six bacteria exposed to both Cu-BTC and NO⊂Cu-BTC ranged from 47.858 to 803.39 mm^2^, and the zone was relative smaller for *S. aureus,* compared to that for the other five bacteria exposed to NO⊂Cu-BTC. In contrast, the inhibition zone for *S. aureus* exposed to Cu-BTC was 376.84 mm^2^, which was larger than that of *S. aureus* exposed to NO⊂Cu-BTC (Table 3). The NO-releasing Cu-BTC was more effective in inhibiting the growth of the five tested bacteria than the growth of *S. aureus.* Furthermore, the data for the *P. aeruginosa* strains, which displayed the least susceptibility, and *E. coli*, which displayed the greatest susceptibility, were consistent with those in previous reports [47,55]. The bacterial species were determined as being susceptible to NO⊂Cu-BTC in the following order: *E. coli* > MRSA (clinically isolated) > *S. aureus* > MRSA (KCCM 40510) > *B. cereus* > *P. aeruginosa*. Thus, NO⊂Cu-BTC exhibited more selective and synergistic antimicrobial activities toward a wide range of bacteria than Cu-BTC.

## 3. Materials and Methods

### 3.1. Preparation of Cu-BTC **1** and **2**

The Micro-sized Cu-BTC (**1**) was prepared according to a reported protocol with slight modifications [45]. In a typical synthesis, copper(II) nitrate trihydrate (Cu(NO_3_)_2_∙3H_2_O, 99%, Acros, Seoul, Korea) (0.725 g, 3.0 mmol) was dissolved in 10 mL of deionized water. In a separate vial, 1,3,5-benzenetricarboxylic acid (H_3_BTC, C_9_H_6_O_6_, 98%, Acros, Seoul, Korea) (0.210 g, 1.0 mmol) was dissolved in 10 mL of ethanol. To a solution of H_3_BTC stirring at 500 rpm for 10 min at room temperature, the Cu(NO_3_)_2_ solution was quickly added. Then, 0.7 mL of DMF was added to the solution. The reaction mixture was placed in a temperature-controlled oven at 80 °C for 24 h. After cooling to 25 °C, the product was collected by centrifugation, washed several times with solvents (deionized water and ethanol), and dried under vacuum for 24 h before further use.

Nano-sized Cu-BTC (**2**) was prepared from **1** by ball-milling at 350 rpm for 6 h with a balls-to-Cu-BTC weight ratio of 50:1 using the Fritsch™ Planetary Mill Pulverisette 5 (Idar-Oberstein, Germany). Powder X-ray diffraction (PXRD), FTIR, and scanning-electron microscopy (SEM) were used for further characterization of the Cu-BTC MOFs.

### 3.2. X-ray Crystallography

X-ray diffraction data for NO⊂Cu-BTC were collected on a Bruker APX-II diffractometer (Ames, IA, USA) equipped with a monochromator with a Mo Kα (λ = 0.71073 Å) incident beam at the National Research Facilities and Equipment Center (NanoBio-Energy Materials Center) at Ewha Womans University. A crystal was mounted on a glass fiber. The CCD data were integrated and scaled using the Bruker-SAINT software package (Bruker Nano, Inc., 2019), and the structure was solved and refined using SHEXL-2014 (Sheldrick, 2014) [58]. All hydrogen atoms were placed at the calculated positions. The crystallographic data are listed in Table 1, and the bond lengths and angles are listed in Appendix A. The CCDC reference number is 2073930 for NO⊂Cu-BTC.

### 3.3. NO Storage/Release of Cu-BTC

For the dehydration process, 10 mg of **1** and **2** were dehydrated in a vacuum at 150 °C for 24 h. The activated Cu-BTCs were placed in a 40 mL vial, and the vial was placed in a Parr bottle (200 mL) that was connected to an in-house NO reactor. The reactor was flushed with Ar gas (99.99%) thrice for 10 min each to remove oxygen before charging NO. The reaction bottle was then charged under 10 atm of NO using ultrapure grade (99.5%) NO gas provided by Dong-A Specialty Gases (Seoul, Korea), which was purified over KOH pellets to remove trace NO degradation products. The bottle was then sealed for 72 h at 25 °C. Prior to removing the NO-charged Cu-BTCs, unreacted NO was purged from the chamber with Ar thrice. The NO⊂Cu-BTCs (**3** and **4**) were stored in a sealed container at −20 °C until further use. The NO release profiles of NO⊂Cu-BTC were monitored in deoxygenated phosphate-buffered saline (PBS; 0.01 M, pH 7.4) at 37 °C using a Sievers 280i chemiluminescence NO analyzer (Boulder, CO, USA). NO released from NO⊂Cu-BTC was transported to the analyzer by a stream of Ar gas (70 mL·min^−1^) passed through the reaction cell. The instrument was calibrated with air passed through a zero filter (0 ppm NO) and 45 ppm of NO standard gas balanced with N_2_ purchased from Dong-Woo Gas Tech (Siheung, Korea). The total amount of NO released, t[NO]; maximum flux of NO release, [NO]_m_; time necessary to reach [NO]_m_, *t*_m_; half-life of NO release, *t*_1/2_; and duration time of NO release for sustained fluxes of NO ≥ 1 ppb·mg^−1^, *t*_d_, were determined for the evaluation of NO⊂Cu-BTC.

### 3.4. Antibacterial Test

To confirm the antibacterial activity of the two types of Cu-BTC, we used six strains of bacteria, which included two drug-resistant bacteria. Two Gram-negative strains (*Escherichia coli* ATCC 11775 and *Pseudomonas aeruginosa* ATCC 9027) and two Gram-positive strains (*S. aureus* ATCC 14458 and *B. cereus* ATCC11706) were acquired from the American Type Culture Collection (ATCC, Rockville, MD, USA). One methicillin-resistant bacteria, *S. aureus* (MRSA KCCM 40510), was purchased from the Korean Culture Center of Microorganisms (KCCM, Sedaemun-Gu, Seoul, Korea), and clinically isolated MRSA was kindly provided by the Yonsei Medical Center in Seoul. Gram-negative and Gram-positive bacteria were grown on a plate count agar (PCA, Becton, Dickinson and Company, Sparks, MD, USA) at 35 °C for 24 h. MRSA strains were grown on Brain Heart Infusion agar (BHIA, Becton, Dickinson and Company) at 35 °C for 24 h. The antibacterial activities of Cu-BTC and NO⊂Cu-BTC were evaluated using a disk diffusion method from the National Committee for Clinical Laboratory Standards (NCCLS 2003a), protocol M2-A8. A bacterial suspension for each strain was prepared by the resuspension of each bacterial colony in sterilized 0.85% saline solution, and the optical density of the suspension solution was adjusted to 0.5 McFarland turbidity standards (~10^7^ CFU/mL). Each bacterial suspension was applied to the entire surface of the PCA and BHIA plates, and the agar plates were incubated for 10 min at room temperature. Next, three 10 mm Whatman filter paper disks were placed on each agar plate, as shown in Figure 1, and each suspension solution (final concentration: 1 mg/mL in 0.85% saline solution) of Cu-BTC and NO⊂Cu-BTC was applied to the paper disk. The agar plates treated with each sample were incubated at 35 °C for 24 to 48 h, and the zone of inhibition of each agar plate was measured by subtracting the diameter of each disk from the diameter of the total inhibition zone using ImageJ software (NIH, Bethesda, MD, USA). All experiments were carried out in triplicate with results expressed as a mean with standard deviations (S.D). Statistical comparisons between Cu-BTC and NO⊂Cu-BTC performed using Stutens’s *t*-test and a difference with *p* < 0.05 was considered statistically significant.

### 3.5. Instrumentation

The PXRD patterns of the Cu-BTCs were recorded using a Rigaku MiniFlex diffractometer (Rigaku Corp., Neu-Isenburg, Germany). FTIR spectra were measured on a Nicolet iS10 FTIR spectrometer with KBr pellets (Thermo Fisher Scientific, Waltham, MA, USA). TGA was performed using a TG 209 F3 Tarsus^®^ instrument (NETZSCH, Burlington, MA, USA). The surface morphology and elemental composition of Cu-BTC were characterized using SEM-EDS (FE-SEM, JEOL JSM-5800F, Peabody, MA, USA). N_2_ adsorption isotherms were obtained by using a BELSORP-mini II instrument (BEL Japan, Inc., Tokyo, Japan). High-purity (99.999%) gases were used throughout the adsorption experiments. All samples were activated by rinsing them thoroughly, followed by drying under vacuum for 24 h prior to the gas sorption measurements.

## 4. Conclusions

Microporous Cu-BTC MOFs with two different morphologies (micro-sized and nano-sized) were prepared, and their NO sorption/release properties were investigated. NO was captured in Cu-BTC by partial hydrogen bonding with water molecules coordinated to Cu and ON**^…^**O interactions with the oxygen atoms of the BTC linker. The solid-state structural analysis of Cu-BTC revealed a stable structure upon NO loading and releasing. When the nano-sized Cu-BTC MOF was activated, the total amount of NO released was significantly higher than that released from the micro-sized Cu-BTC, owing to the lower hydration and different morphological characteristics of the former. Furthermore, the NO⊂Cu-BTC MOFs showed bactericidal properties against five strains of bacteria that were superior to those of the Cu-BTC MOFs. These results reveal that the NO release of Cu-BTC can be highly enhanced by controlling the hydration degree and morphology of the MOF, and the resultant NO delivery vehicle system exhibits synergistic therapeutic effects. This study paves the way for the development of various MOFs as promising hybrid NO donor materials through simple physical modifications for application as drug delivery systems.

## Figures and Tables

**Figure 1 ijms-23-09098-f001:**
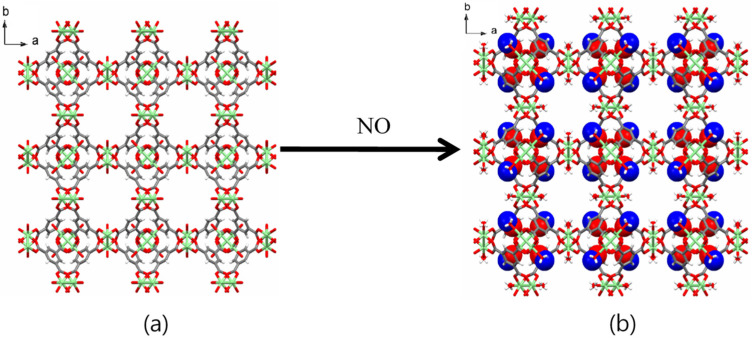
(**a**) Structure of Cu-BTC, *Fm-3m*, a = b = c = 26.3015(4) Å, V = 18194.6(5) Å^3^. (**b**) Structure of NO⊂micro-Cu-BTC, *Fm-3m*, a = b = c = 26.3068(11) Å, V = 18205.6(5) Å^3^. NO molecules are shown in space filled models. The color codes: green, Cu; grey, carbon; red, oxygen; blue, nitrogen.

**Figure 2 ijms-23-09098-f002:**
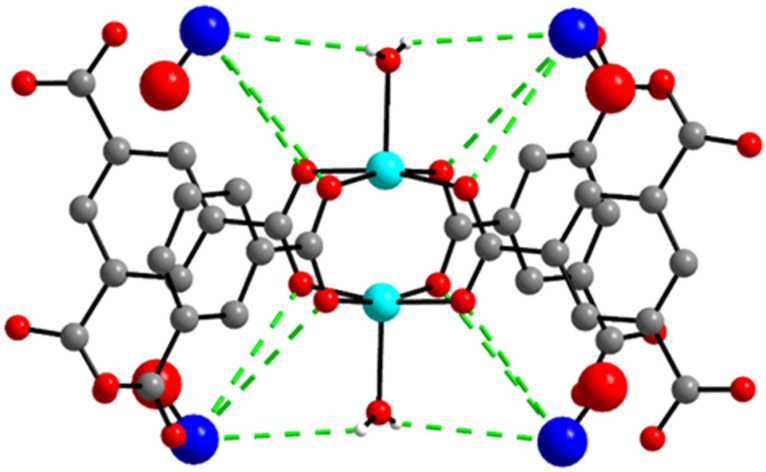
Fragment structure showing weak hydrogen bonding interactions between NO and the coordinated water molecules, and interactions between NO and carboxylate oxygen atoms. Color codes: green, Cu; grey, carbon; red, oxygen; blue, nitrogen. Hydrogen atoms, except those of water, were omitted. The large spheres represent NO for clarity.

**Figure 3 ijms-23-09098-f003:**
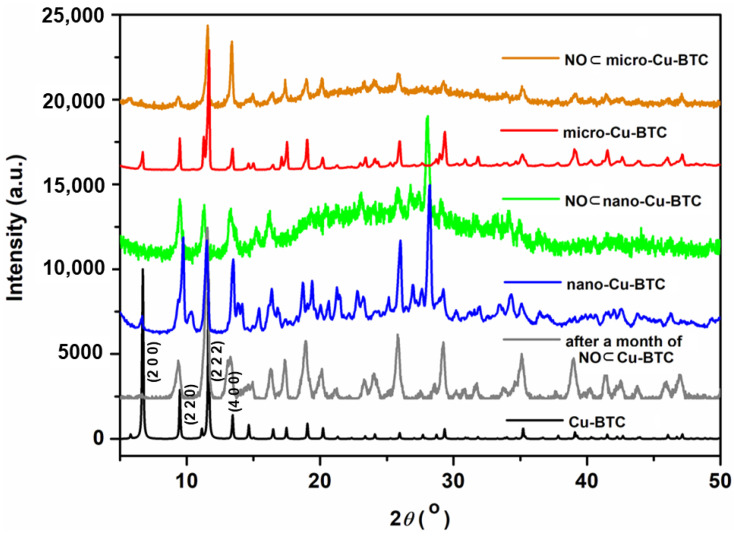
Powder X-ray diffraction (PXRD) of micro-sized Cu-BTC **1** (red)**,** nano-sized Cu-BTC **2** (blue), NO⊂micro-Cu-BTC **3** (yellow), NO⊂nano-Cu-BTC **4** (green), and NO⊂micro-Cu-BTC **3** after loading NO (grey).

**Figure 4 ijms-23-09098-f004:**
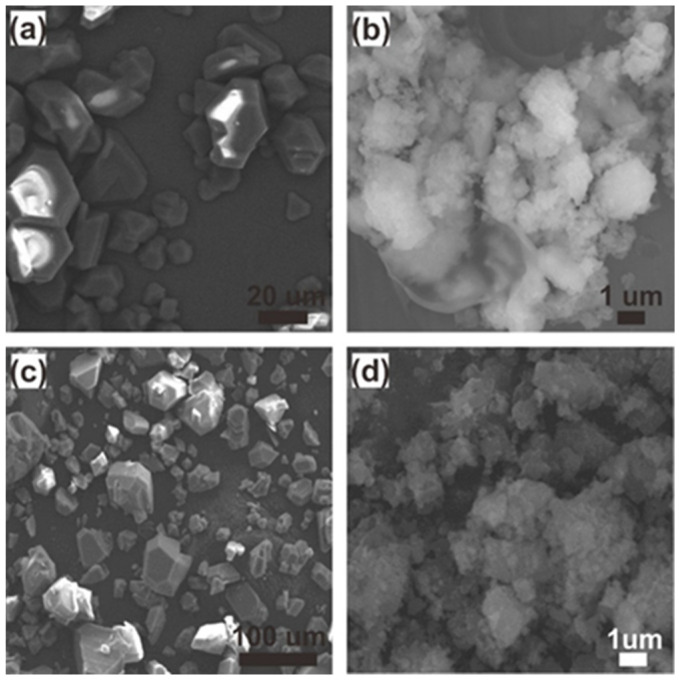
Scanning electron microscopy (SEM) images of (**a**) micro-sized Cu-BTC (**1**), (**b**) nano-sized Cu-BTC (**2**), (**c**) NO⊂micro-Cu-BTC (**3**), and (**d**) NO⊂nano-Cu-BTC (**4**).

**Figure 5 ijms-23-09098-f005:**
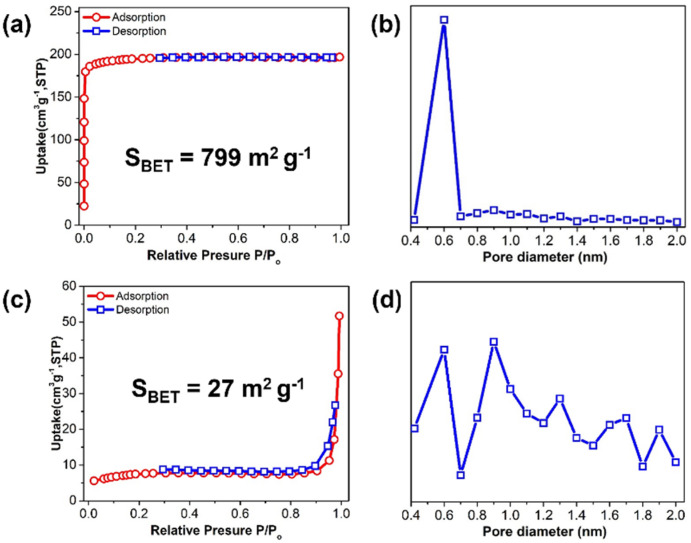
N_2_ sorption, measured at 77 K, and pore size of **1** (**a**,**b**) and **2** (**c**,**d**).

**Figure 6 ijms-23-09098-f006:**
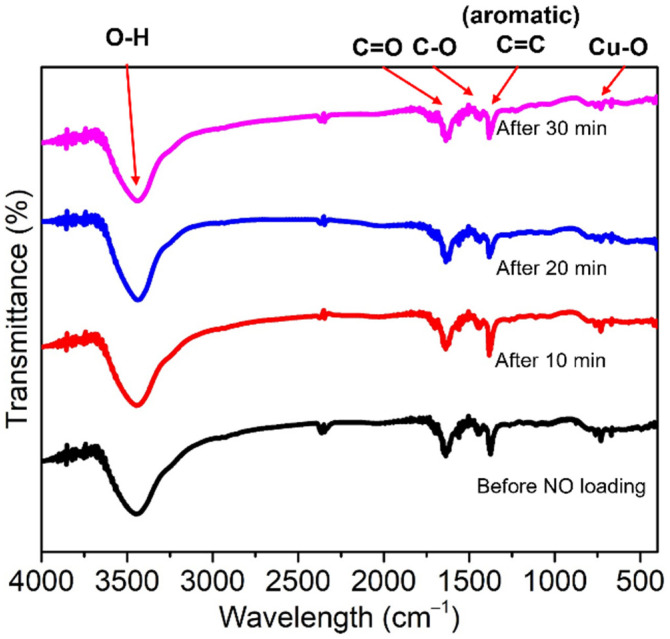
FTIR spectra of **1** (black) and **3** after releasing NO for 10, 20, and 30 min (red, blue, and pink, respectively).

**Figure 7 ijms-23-09098-f007:**
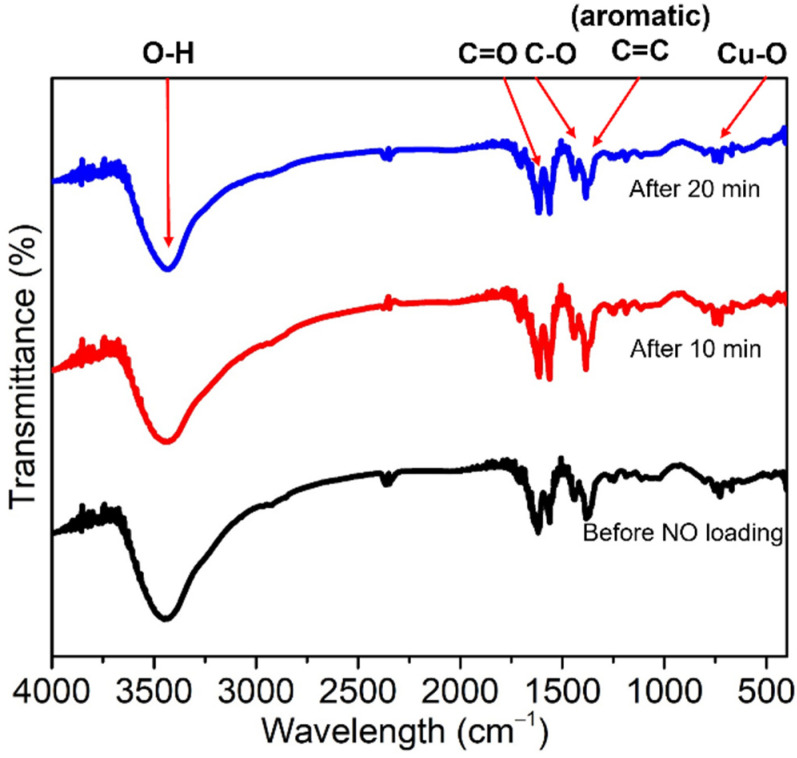
FTIR spectra of **2** (black) and **4** after releasing NO for 10 and 20 min (red and blue, respectively).

**Figure 8 ijms-23-09098-f008:**
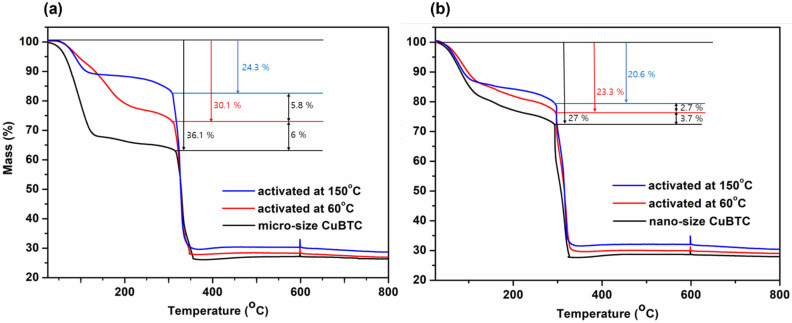
TGA of **1** (**a**) and **2** (**b**)**.** Black line, as-prepared; red line, activated at 60 °C; blue line, activated at 150 °C.

**Figure 9 ijms-23-09098-f009:**
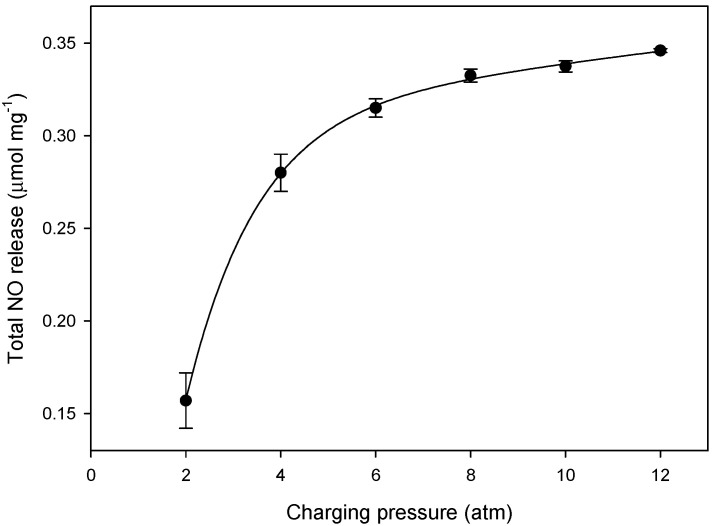
Plot of total NO release of NO⊂micro-Cu-BTC **3** in deoxygenated phosphate-buffered saline (PBS, 0.01 M, pH 7.4) at 37 °C as a function of varying NO charging pressure.

**Figure 10 ijms-23-09098-f010:**
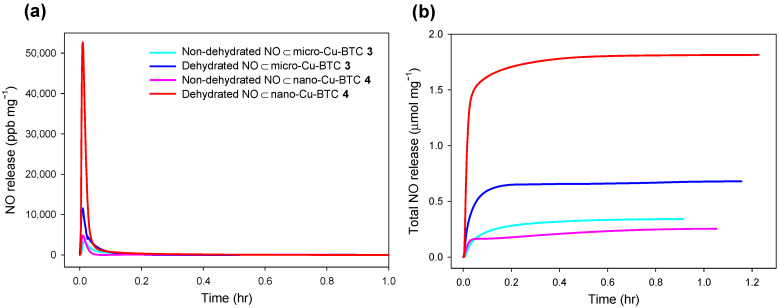
(**a**) Real-time NO release profiles and (**b**) total amount of NO released for NO⊂Cu-BTC in deoxygenated PBS (0.01 M, pH 7.4) at 37 °C: sky-blue line, non-dehydrated NO⊂micro-Cu-BTC **3**; blue line, dehydrated NO⊂micro-Cu-BTC **3**; pink line, non-dehydrated NO⊂nano-Cu-BTC 4; red line, dehydrated NO⊂nano-Cu-BTC **4**.

**Figure 11 ijms-23-09098-f011:**
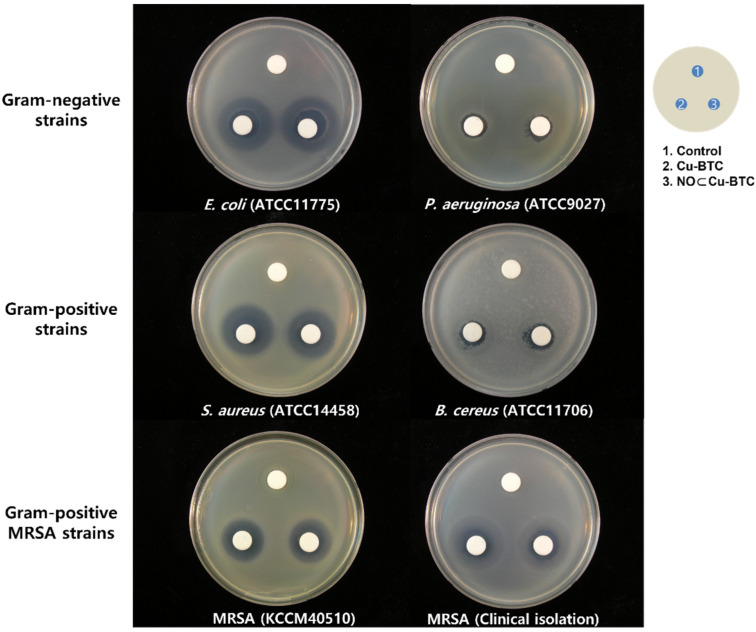
Inhibition zone assay of Cu-BTC (BTC: benzene-1,3,5-tricarboxylate) and NO⊂Cu-BTC (NO: nitric oxide) toward six strains of bacteria tested by disc diffusion method.

**Table 1 ijms-23-09098-t001:** Crystallographic data for NO⊂Cu-BTC.

Empirical Formula	C_6_H_2_CuO_5_ (NO)_0.17_
Formula weight	222.62
Temp. (K)	223(2)
Wavelength (Å)	0.71073
Space group	*Fm-3m*
a (Å)	26.3068(11)
b (Å)	26.3068(11)
c (Å)	26.3068(11)
α (°)	90.00
β (°)	90.00
γ (°)	90.00
Volume (Å^3^)	18,206(2)
Z	48
Density (calc.) (Mg m^−3^)	0.975
Absorption coeff. (mm^−1^)	1.429
Crystal size (mm)	0.090 × 0.080 × 0.020
Reflections collected	148,341
Independent reflections	1177 [R(int) = 0.1842]
Data/restraints/parameters	1177/1/40
Goodness-of-fit on F^2^	1.291
Final R indices [I>2σ(I)]	R1 = 0.1149, wR2 = 0.2110
R indices (all data)	R1 = 0.1264, wR2 = 0.2170
Largest diff. peak and hole (e.Å-3)	0.697 and −0.564
CCDC	2,073,930

**Table 2 ijms-23-09098-t002:** NO release properties of NO⊂Cu-BTC ^a,b^.

	t[NO] (μmol·mg^−1^)		*t*_m_ (s)		[NO]_m_ (ppm·mg^−1^)		*t*_1/2_ (s)		*t*_d_ (h)	
	3	4	3	4	3	4	3	4	3	4
Non-dehydrated	0.34	0.25	50	40	3.25	4.78	185	75	0.9	1.1
Dehydrated	0.68	1.81	33	39	11.5	52.7	73	58	1.2	1.2

^a^ Values were determined using a Sievers chemiluminescence NO analyzer (NOA 280i) in deoxygenated PBS (0.01 M, pH 7.4) at 37 °C. ^b^ t[NO], total amount of NO released; [NO]_m_, maximum flux of NO release; *t*_m_, time necessary to reach [NO]_m_; *t*_1/2_, half-life of NO release; *t*_d_, duration time of NO release for sustained fluxes of NO ≥ 1 ppb⋅mg^−1^.

**Table 3 ijms-23-09098-t003:** Inhibition zone of Cu-BTC and NO⊂Cu-BTC toward six strains of bacteria.

Bacterial Strains *		Inhibition Zone (mm^2^)		*p* Value
		Cu-BTC	NO⊂Cu-BTC	
Gram-negative strains	*E. coli* (ATCC11775)	712.3 ± 38.6	803.4 ± 54.7	<0.024
	*P. aeruginosa* (ATCC9027)	47.8 ± 22.2	52.3 ± 26.2	<0.21
Gram-positive strains	*S. aureus* (ATCC14458)	414.2 ± 18.4	376.8 ± 21.6	<0.016
	*B. cereus* (ATCC11706)	51.5 ± 7.3	74 ± 8.8	<0.003
Gram-positive MRSA strains	MRSA (KCCM40510)	271.4 ± 8.1	297.4 ± 8.1	<0.0018
	MRSA (Clinical isolation)	501.1 ± 5.6	573.3 ± 6.9	<0.00057

* Escherichia coli (*E. coli* ATCC11775); Pseudomonas aeruginosa (*P. aeruginosa* ATCC9027)); Staphylococcus aureus (*S. aureus* ATCC14458); Bacillus cereus (*B. cereus* ATCC11706)); methicillin-resistant Staphylococcus aureus strains (MRSA KCCM40510 and clinically isolated MRSA). Data are mean zone inhibition (mm^2^) ± standard deviation (S.D.) of three replicates (*n* = 3, *p* < 0.05).

## Data Availability

Not applicable.

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
