# Peer review of "Controllable Nitric Oxide Storage and Release in Cu-BTC: Crystallographic Insights and Bioactivity"

_ijms, 2022, doi:10.3390/ijms23169098_

Round 1
Reviewer 1 Report
The manuscript by Lee et al. deals with the synthesis of microporous Cu-BTC MOFs with two different morphologies (micro-sized and nano-sized), prepared by a modified solvothermal method, for improving its NO storage/release capacity and antibacterial activity. The paper is well written, and the discussion of results is convincing. I only have minor remarks, listed below.
1. Please check the completeness of the sentence on lines 83-84.
2. Please consider adding the error bars in Figures 9 and 10; moreover, Figure 10 is poorly resolved and the character size is tiny, making the readability difficult.
3. In Table 3, revise the reliability of the inhibition zone's mean according to the reported error.
Author Response
Comments and Suggestions for Authors
The manuscript by Lee et al. deals with the synthesis of microporous Cu-BTC MOFs with two different morphologies (micro-sized and nano-sized), prepared by a modified solvothermal method, for improving its NO storage/release capacity and antibacterial activity. The paper is well written, and the discussion of results is convincing. I only have minor remarks, listed below.
- Please check the completeness of the sentence on lines 83-84.
Response: Thank you for your valuable comment, sentences on lines 83-82 was revised to ’Herein, for improving the biocompatibility and NO capacity, nano-sized Cu-BTC was fabricated via a convenient and straightforward ball-milling process of micro-sized Cu-BTC.
- Please consider adding the error bars in Figures 9 and 10; moreover, Figure 10 is poorly resolved and the character size is tiny, making the readability difficult.
Response: Thank you for your comment, we added error bars in Figure 9. Figure 10 is resolved to be easily readable as followings.
Figure 9. Plot of total NO release of NO⊂micro-Cu-BTC 3 in deoxygenated phosphate-buffered saline (PBS, 0.01 M, pH 7.4) at 37 °C as a function of varying NO charging pressure.
Figure 10. (a) Real-time NO release profiles and (b) total amount of NO released for NO⊂Cu-BTC in deoxygenated PBS (0.01 M, pH 7.4) at 37 °C: sky-blue line, non-dehydrated NO⊂micro-Cu-BTC 3; blue line, dehydrated NO⊂micro-Cu-BTC 3; pink line, non-dehydrated NO⊂nano-Cu-BTC 4; red line, dehydrated NO⊂nano-Cu-BTC 4.
- In Table 3, revise the reliability of the inhibition zone's mean according to the reported error.
Response: As you pointed out, we revised our manuscript by adding statistical analysis data in Table 3 and by adding the statistical analysis method in materials and methods part.
Table 3. Inhibition zone of Cu-BTC and NO⊂Cu-BTC toward six strains of bacteria.
|
Bacterial Strains* |
Inhibition Zone (mm2) |
p value |
||
|
Cu-BTC |
NO⊂Cu-BTC |
|||
|
Gram-negative strains |
E. coli (ATCC11775) |
712.3±38.6 |
803.4±54.7 |
<0.024 |
|
P. aeruginosa (ATCC9027) |
47.8±22.2 |
52.3±26.2 |
<0.21 |
|
|
Gram-positive strains |
S. aureus (ATCC14458) |
414.2±18.4 |
376.8±21.6 |
<0.016 |
|
B. cereus (ATCC11706) |
51.5±7.3 |
74±8.8 |
<0.003 |
|
|
Gram-positive MRSA strains |
MRSA (KCCM40510) |
271.4±8.1 |
297.4±8.1 |
<0.0018 |
|
MRSA (Clinical isolation) |
501.1±5.6 |
573.3±6.9 |
<0.00057 |
|
* Escherichia coli (E. coli ATCC11775); Pseudomonas aeruginosa (P. aeruginosa ATCC9027)); Staphylococcus aureus (S. aureus ATCC14458); Bacillus cereus (B. cereus ATCC11706)); methicillin-resistant Staphylococcus aureus strains (MRSA KCCM40510 and clinically isolated MRSA). Data are mean zone inhibition (mm2) ± standard deviation (S.D) of three replicates (n=3, p <0.05).

Reviewer 2 Report
The utilization of MOF-based hybrid materials in innovative biomedicine applications is “hot spot” in the material chemistry. So, the paper by D.N. Lee reported the investigation of the peculiarities of NO storage and release in the micro/nano Cu-BTC materials by structural examinations is interesting from this point of view.
The paper may be published after minor revision by addressing the following points.
1. Please discuss the differences in XRD patterns for the synthesized materials (Fig. 3). Please present XRD patterns for pristine microBTC and nanoBTC materials.
2. The specific surface area of 799 m2/g measured for microBTC sample is no high, it is low at least as twice-fold as compared reported HKUST-1 samples. What is the reason for this specific surface area decrease, phase impurities?
3. What is outer surface and micro/meso pore volume for the synthesized materials?
4. The low specific surface for the nanoBTC material is due to is partial destruction after ball-milling. This suggestion is confirmed by XRD results, which show the significant amorphization of this material.
5. The relevant review doi: 10.3389/fbioe.2021.60360 should be cited in the paper.
Author Response
Comments and Suggestions for Authors
The utilization of MOF-based hybrid materials in innovative biomedicine applications is “hot spot” in the material chemistry. So, the paper by D.N. Lee reported the investigation of the peculiarities of NO storage and release in the micro/nano Cu-BTC materials by structural examinations is interesting from this point of view.
The paper may be published after minor revision by addressing the following points.
- Please discuss the differences in XRD patterns for the synthesized materials (Fig. 3). Please present XRD patterns for pristine microBTC and nanoBTC materials..
Response: Thank you for your valuable comment, we present XRD patterns for pristine micro-Cu-BTC and nano-Cu-BTC materials on new Fig.3. We added the sentence as followings.
Figure 3 shows the XRD patterns of the simulated and NO-loaded Cu-BTC MOFs of different sizes. All XRD patterns coincided well with the simulated pattern. The main peaks at 6.82°, 9.64°, 11.76° and 13.57° corresponding (2 0 0), (2 2 0), (2 2 2) and (4 0 0) of the Cu-BTC pattern were maintained in nano-Cu-BTC and micro-Cu-BTC. The (2 0 0) peak was not shown in NO loaded Cu-BTC MOFs. Particularly, the PXRD of 4 obtained after storing NO for 1 month did not show any structural changes, and the original framework of Cu-BTC remained unchanged.
Figure 3. Powder X-ray diffraction (PXRD) of micro-sized Cu-BTC 1 (red), nano-sized Cu-BTC 2 (blue), NO⊂micro-Cu-BTC 3 (yellow), NO⊂nano-Cu-BTC 4 (green), and NO⊂micro-Cu-BTC 3 after loading NO (grey).
- The specific surface area of 799 m2/g measured for microBTC sample is no high, it is low at least as twice-fold as compared reported HKUST-1 samples. What is the reason for this specific surface area decrease, phase impurities?
Response: We agree with the reviewer that the current specific surface area of micro-sized Cu-BTC (799 m2 g-1) is not as high as that of the reported data, e.g., 1740 m2 g-1 (J. Am. Chem. Soc., 2015, 137, 10009−10015), 876.5 m2 g-1 (Chem. Eng. J., 2022, 430, 133088). However, there also has Cu-BTC in which their specific surface area was lower than the current micro-sized Cu-BTC, for example, 692.2 m2 g-1 (Science, 1999, 283, 1148–1150).
To prepare micro-sized Cu-BTC, we followed the previous reported procedure (J. Am. Chem.Soc., 2012, 134, 51−54) without any modification. Unfortunately, in the reference, the specific surface area of Cu-BTC was not performed, therefore, we could not compare with our specific surface area value.
The decreased specific surface area of micro-sized Cu-BTC could be originated by following reasons. First, micro-sized Cu-BTC could still contain both ethanol and water as coordinated molecules because these two solvents were used in the synthetic procedure. Second, the specific surface area of sample also depends on the experimental conditions for nitrogen sorption measurement, for instance, temperature and time for evacuation step. In this work, the micro-sized Cu-BTC was evacuated at 25 °C for 12 h before checking. Third, the particle size of micro-sized Cu-BTC is quite big (several tens of micrometer), so it might affect to the specific surface area.
- What is outer surface and micro/meso pore volume for the synthesized materials?
Response: Thank you for your important comment, we have summarized the nitrogen sorption properties of micro-sized Cu-BTC and nano-sized Cu-BTC on Table S2.
Table S2. The nitrogen sorption properties of micro-sized Cu-BTC and nano-sized Cu-BTC.
|
|
||
|
|
Micro-sized Cu-BTC |
Nano-sized Cu-BTC |
|
Specific surface area (m2 g-1) |
799 |
27 |
|
Micropore volume (cm3 g-1) |
0.3283 |
0.0092 |
|
Mesopore volume (cm3 g-1) |
0.0096 |
0.0570 |
|
Mean pore diameter (nm) |
1.5234 |
9.9172 |
Unlucky, we could not determine the outer surface of the two Cu-BTC samples.
- The low specific surface for the nano BTC material is due to is partial destruction after ball-milling. This suggestion is confirmed by XRD results, which show the significant amorphization of this material.
Response: Both nano-sized Cu-BTC 2 and NO⊂nano-Cu-BTC 4 showed new appearing peak near 28° probably attributed to amorphization.
- The relevant review doi: 10.3389/fbioe.2021.60360 should be cited in the paper.
Response: We can not find this DOI #. If more publishing informations such as title and author are given on the relevant review, I will cite it.